# Intracerebral Plasmacytoma in a Patient with HIV-1 Infection and SARS-CoV-2 Superinfection

**DOI:** 10.3390/brainsci12060705

**Published:** 2022-05-30

**Authors:** Jurijs Nazarovs, Daira Lapse, Gunta Stūre, Marina Soloveičika, Līga Jaunozolina, Elīna Ozoliņa, Sandra Lejniece

**Affiliations:** 1Pathology Centre, Riga East Clinical University Hospital, Hipokrāta Street 2, LV-1038 Rīga, Latvia; daira.lapse@stradini.lv (D.L.); liga.jaunozolina@rsu.lv (L.J.); elina.ozolina@aslimnica.lv (E.O.); 2Pathology Institute, Pauls Stradins Clinical University Hospital, Pilsoņu Street 13, LV-1002 Rīga, Latvia; 3Department of Pathology, Riga Stradins University, Dzirciema Street 16, LV-1007 Rīga, Latvia; 4Latvian Centre of Infectious Diseases, Riga East Clinical University Hospital, Linezera, LV-1002 Rīga, Latvia; gunta.sture@rsu.lv; 5Department of Infectious Diseases, Riga Stradins University, Dzirciema Street 16, LV-1007 Rīga, Latvia; 6Department of Laboratory, Riga East Clinical University Hospital, Hipokrata Street 2, LV-1038 Rīga, Latvia; marina.soloveicika@aslimnica.lv; 7Department of Radiology, Riga Stradins University, Dzirciema Street 16, LV-1007 Rīga, Latvia; 8Hematology and Chemotherapy Clinic, Riga East Clinical University Hospital, Hipokrāta Street 2, LV-1038 Rīga, Latvia; sandra.lejniece@rsu.lv; 9Department of Internal Diseases, Riga Stradins University, Dzirciema Street 16, LV-1007 Rīga, Latvia

**Keywords:** plasmacytoma, rare intracerebral tumor, intracranial plasmacytoma

## Abstract

We present a rare case of intracranial solitary plasmacytoma arising in brain parenchyma in the basal nuclei. Clinical management and autopsy results of the case are described. Background: Intracranial plasmacytomas arising from brain parenchyma are extremely rare, and data from the literature are limited. Primary intracranial plasmacytomas are rare because plasma cells are not found in the brain in normal conditions. Commonly, intracranial plasmacytoma is associated with multiple myeloma, which is why multiple myeloma must be ruled out to diagnose solitary intracranial plasmacytoma. Considering that solitary plasmacytoma and multiple myeloma have some histopathological similarities, it is important to differentiate them because their respective treatments and prognoses are different. Imaging features of primary extramedullary plasmacytoma are nonspecific but are compatible with solid tumors with invariable enhancement. Plasmacytoma was aggressive because it was not diagnosed after the first MRI, but 1.5 months later, MRI showed a large object. We present a rare case of intracranial solitary plasmacytoma arising in brain parenchyma in the basal nuclei.

## 1. Introduction

The first HIV patient in Latvia was registered in 1987. In Latvia, 63% of all registered patients were late HIV presenters. Patients with severe immunodeficiencies have a high risk of becoming sick with many opportunistic and oncological diseases simultaneously, such as cerebral toxoplasmosis, tuberculosis, lymphoma, and cryptococcosis, making it more difficult to confirm the diagnosis and extend mortality.

Extraosseous plasmacytomas are plasma cell tumors that comprise about 1% of all plasma cell neoplasms. Most commonly, they occur in the mucous membranes of the upper airways [1,2,3]. Extraosseous plasmacytomas can arise in different anatomic sites, including the gastrointestinal tract, skin, thyroid gland, CNS, parotid gland, lymph nodes, breast, bladder, and testes [4]. Solitary intracranial plasmacytomas can involve the skull, meninges, or brain [2,5]. Intracranial plasmacytomas arising from the brain parenchyma are extremely rare, and data from the literature are limited. In normal conditions, plasma cells are not found in the brain [1,6].

Intracranial plasmacytoma generally involves the meninges or the skull, mostly with dural attachment, and just a few cases have been reported with isolated intraparenchymal disease [7]. CT is superior to MRI in delineating subtle bone involvement, but MRI provides excellent anatomical detail and allows for soft tissue characterization. Imaging features of primary extramedullary plasmacytoma are nonspecific but are compatible with solid tumors with invariable enhancement. Large tumors may show areas with destruction, infiltration, or encasement of adjacent structures, but imaging alone cannot differentiate these tumors from more common malignant entities [8].

We report a case of a solitary primary extraosseous plasmacytoma of the brain located in the right frontal and temporal lobe with invasion in lateral ventricles.

## 2. Results

A 30-year-old woman was hospitalized at Riga East Clinical University hospital in March of 2020 with long-time high temperatures, diarrhea, and weight loss. When the patient arrived at the hospital, no registered neurological symptoms were found, and the first CT of the brain did not uncover any pathologies. This was her first HIV examination, and she was confirmed to have HIV infection, stage C III. Upon admission, her CD4 count was 32, but when antiretroviral therapy was started, her CD4 cell count increased to 291.

After a second examination, results confirmed disseminated atypical mycobacteriosis—MAC—in feces, blood, and bronchoalveolar-lavage-cultured M. avium. As a result, the patient started specific antiretroviral and antimycobacterial therapy.

After 1.5 months, she developed a weakness in the left side of the body and left arm and neurocognitive disorders. Serology of opportunistic infection was negative, but magnetic resonance imaging (MRI) of the brain (Figure 1) showed a large lesion in the right caudate nucleus with contrast-enhancing margins (Figure 1C), surrounding vasogenic edema (Figure 1A), and restricted diffusion within the lesion (Figure 1E), respectively.

There were also smaller lesions with vasogenic edema (Figure 1B), restricted diffusion (Figure 1F), and mild contrast enhancement (Figure 1D) in the left caudate nucleus and both parietal lobes. There were no signs of dura mater or skull involvement.

Figure 1 and Figure 2. Magnetic resonance imaging (MRI) of the brain.

Morphologic analysis of cerebrospinal fluid (CSF) showed cytosis—the infiltration of lymphocytes. According to the results of flow cytometry, aq1 monoclonal population of transformed plasmatic cells with the phenotype CD45 + CD19-CD38 + CD138 + CD33 + Kappa + CD117-CD56- was found in the CSF sample (Figure 3). Flow cytometry revealed lymphocytes with an atypical phenotype.

With the primary diagnosis of CNS lymphoma, the patient was assigned a neurosurgeon to decide on a brain biopsy, but it was not performed because the patient was infected with SARS-CoV-2 infection. In September of 2020, the patient was hospitalized with a body temperature of 38.8°C, which lasted for about 3 weeks. Abdominal USG showed specific lymphadenopathy. With a suspected diagnosis of CNS lymphoma, the patient was referred to the Department of Neurosurgery for an interdisciplinary consilium to decide the best course of action. Unfortunately, the patient did not show up to her appointment and did not reach out to reschedule. The patient was hospitalized in the September of 2020 because of a persistent fever of 38.8 °C that lasted for about 3 weeks. Upon this admission, an abdominal USG was performed, which showed specific lymphadenopathy.

Follow-up imaging obtained 6 months after the first imaging (Figure 2) demonstrated a decrease in the size of the large lesion, although the contrast-enhancing area grew (Figure 2C,D). There was also a significant reduction in the vasogenic edema (Figure 2A,B) and the disappearance of lesions in both parietal lobes and the left caudate nucleus, best seen in diffusion-weighted images (Figure 2E,F).

In addition, the patient was diagnosed with oral candidiasis and acute urinary tract infection due to an antibacterial therapy that was prescribed.

A bone marrow trephine biopsy was negative—it showed reactive changes.

The council recommended cerebral toxoplasmosis therapy. Treatment did not result in any positive effects. Based on clinical laboratory data, the hematologist concluded that it could be CNS lymphoma, and due to the patient’s severe general condition, no specific therapy was possible; thus, dexamethasone therapy was suggested. The patient had also contracted COVID-19, and her overall clinical state was rapidly deteriorating. The treatment with dexamethasone provided no effect. A neurosurgical consilium was called, which decided that a brain biopsy, albeit risky due to the involvement of the basal nuclei, was necessary to obtain a definitive diagnosis and decide upon treatment. Unfortunately, it had to be postponed until the patient’s COVID-19 infection had resolved and her general state had improved. The neurosurgeon concluded that the risk of a brain biopsy or operation was too high, and surgery was not recommended. Treatment with dexamethasone was without positive effect. The patient was hospitalized with a further discussion to follow up on the potential of a brain biopsy, which was suspended due to the patient becoming sick with COVID-19. The disease progressed and developed into ex letalis.

An autopsy revealed a brain mass in the parenchyma that was not attached to the meninges. Microscopically, the brain tumor was composed of sheets of non-cohesive cells with rounded nucleus with vesicular/clumped chromatin, prominent nucleoli, and showed perinuclear cytoplasmic clearing and focal necrosis (Figure 4A–D).

Immunohistochemical staining for kappa, lambda, CD138, and LCA was performed. Monoclonality was approved by selective cytoplasmic expression of kappa immunoglobulin light chains (Figure 5A,B). Tumor cells showed strong cytoplasmic positivity for CD138 (Figure 5C).

Bone marrow was normocellular, ~60–75%. The Myeloid cell line was hyperplastic, with delayed maturation. The red blood cell line had a diffuse pattern. M:E = 3–4:1. Blast cells < 2% of the total number of cells. There were rare lymphocytes and plasma cells. Diffuse, small–medium-sized megakaryocytes ranged from 5 to 7 per high-power field (Figure 6).

## 3. Discussion

Fast and accurate verification of the patient’s diagnosis was hampered by a newly diagnosed HIV infection with severe immunodeficiency and multiple diseases at the same time, including MAC, candidiasis, urinary tract infection, plasmacytoma, and COVID-19 infection. The plasmacytoma was aggressive because the first MRI did not diagnose it, but 1.5 months later, MRI showed a large object. The prognosis was complicated by several opportunistic diseases along with diagnostic problems, such as the localization of the plasmacytoma, the limitations of biopsy, the polypharmacy protocol, and the time required to identify M. avium, which reduced co-elasticity during treatment [9].

Solitary intracranial plasmacytoma is a rare tumor, and there are limited descriptions of it in the literature. Primary intracranial plasmacytomas are rare because plasma cells are not found in the brain in usual conditions. Most commonly, intracranial plasmacytoma is associated with multiple myeloma, which is why multiple myeloma must be ruled out to diagnose solitary intracranial plasmacytoma [2,4].

Summarizing the literature, it was concluded that the most common neurological manifestations of SARS CoV-2 infection were acute cerebrovascular diseases, including cerebral hemorrhage, ischemic stroke, cerebral venous sinus thrombosis, and non-specific encephalopathies. These features were distinguished more commonly in older patients and patients with severe disease [10,11].

Among all CNS manifestations, encephalopathy has been reported as the major cause of morbidity and mortality in adult, elderly and sverely ill patients. The presence of comorbidities can further worsen neurological complications and the clinical outcomes.

Patients who have neurodegenerative or inflammation-mediated neurological diseases can also increase the risk of neurologic manifestations in COVID-19 patients [12].

In our patients’ case, SARS CoV-2 infection rapidly worsened her overall clinical state and further investigation, and treatment of the patient was hampered.

The clinical presentation of intracranial plasmacytoma depends on the tumor’s localization and spread. In most cases, it is associated with impaired vision, headache, seizures, or paresis. More often, intracranial plasmacytoma is revealed in the fifth or sixth decade [2]. In our case, the patient was a 30-year-old woman, and she had no symptoms that would indicate an intracerebral lesion.

A brain biopsy would allow a more accurate and faster diagnosis of the tumor. However, due to the detection of SARS-CoV-2 infection, it had to be ruled out. The risks of the operation were also too high.

To establish a diagnosis, a morphological and flow cytometry analysis of CSF was performed. Transformed plasma cells were identified by a specific combination of aberrant expression of surface antigens. The best way to solve this problem is to simultaneously use CD138, CD38, and CD45 markers combined with light-scattering indicators [13]. The exclusion of other differential diagnoses, such as B-cell lymphoma, should be considered. However, in case of B cell lymphoma, lymphocytes on their surface do not contain the CD138 marker.

Laboratory tests showed normal blood tests, and no serum or urinary monoclonal components were found. The bone marrow test was negative. A full skeletal radiological examination was not conducted, but abdominal CT scans were negative.

Histological examination on routine staining is difficult, and immunohistochemistry is very helpful in diagnosing a plasmacytoma. Most important, however, is confirming the monoclonality of the immunoglobulin light chain. CD138 is used for plasma cell staining, but it does not create the diagnosis.

Considering that solitary plasmacytoma and multiple myeloma have some histopathological similarities, it is important to differentiate them because treatment and prognosis are different.

## Figures and Tables

**Figure 1 brainsci-12-00705-f001:**
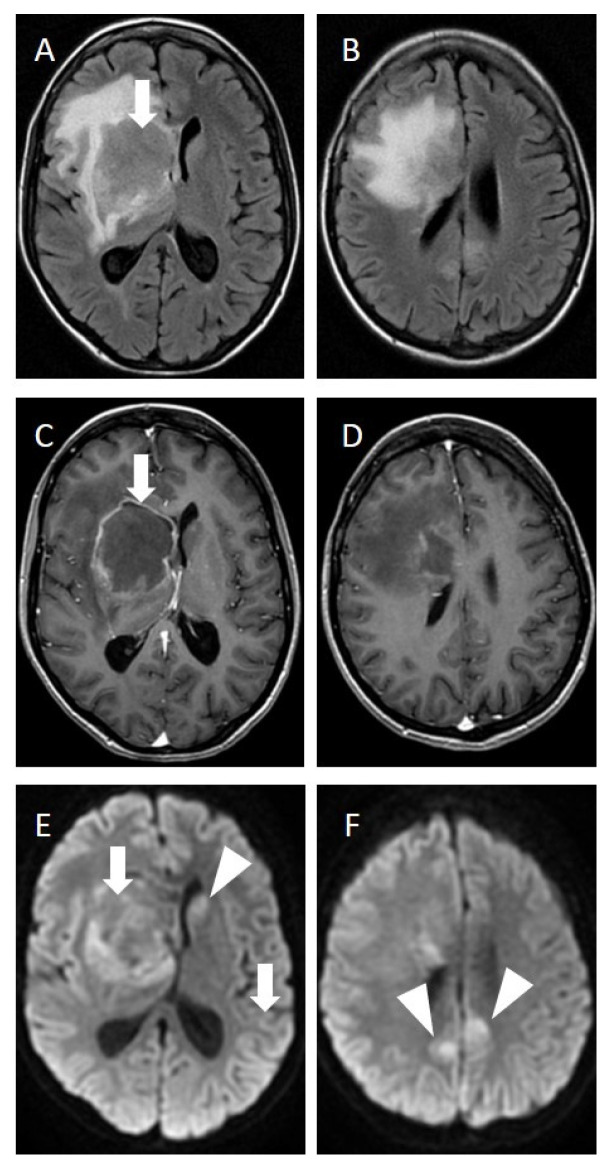
**MRI of the brain.** Large tumour in the right caudate nucleus with contrast enhancing margins (**C**,**D**), surrounding vasogenic edema (**A**,**B**) and restricted diffusion (**E**,**F**) within the lesion (arrow). Small lesions with vasogenic edema, restricted diffusion and contrast enhancement in left caudate nucleus and both parietal lobes (arrowhead).

**Figure 2 brainsci-12-00705-f002:**
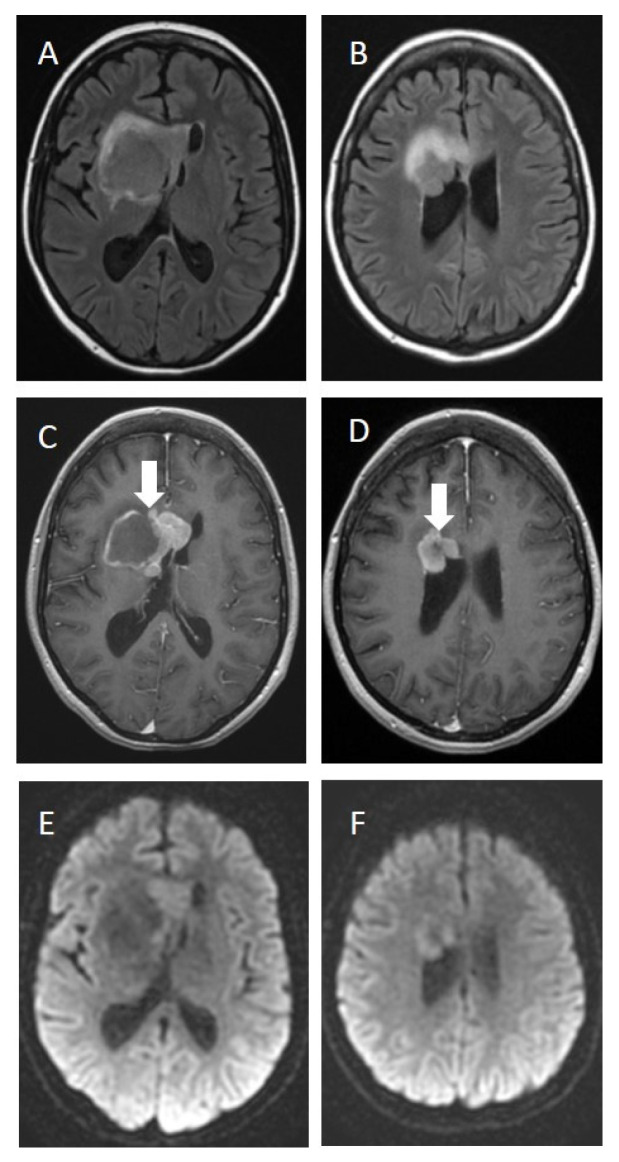
**MRI follow—up imaging after 6 months.** Decrease in the size of the large lesion (**C**,**D**) (arrow), significant reduction in vasogenic edema (**A**,**B**), restricted diffusion (**E**,**F**) and no signs of lesions in both parietal lobes and left caudate nucleus.

**Figure 3 brainsci-12-00705-f003:**
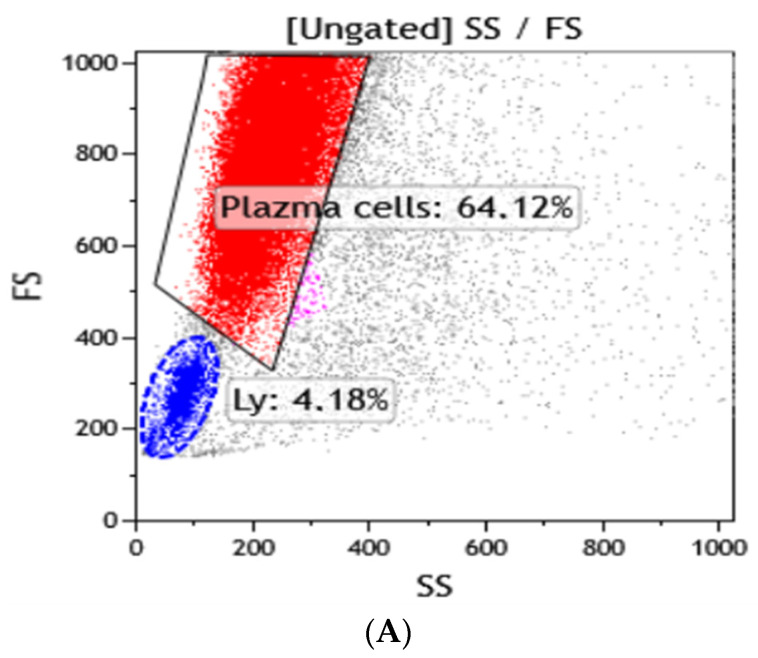
(**A**) CSF sample. Plazma cell population (red color). (**B**) CSF sample. Plazma cell population CD38++ CD138++. (**C**) CSF sample. Monoclonal plazma cell population Kappa++.

**Figure 4 brainsci-12-00705-f004:**
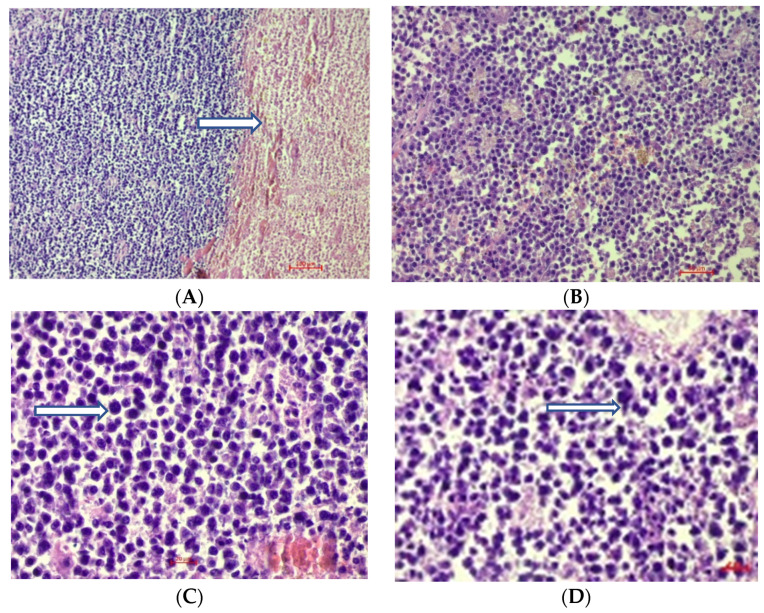
Hematoxylin and eosin staining of formalin-fixed, paraffin-embedded sections of the tumor at: (**A**)—10× magnification with necrotic area (arrow); (**B**)—20× magnification; (**C**,**D**)—40× magnification, showing the proliferation of plasma cells with pleomorphism and enlarged nuclei with clear cytoplasms (arrow).

**Figure 5 brainsci-12-00705-f005:**
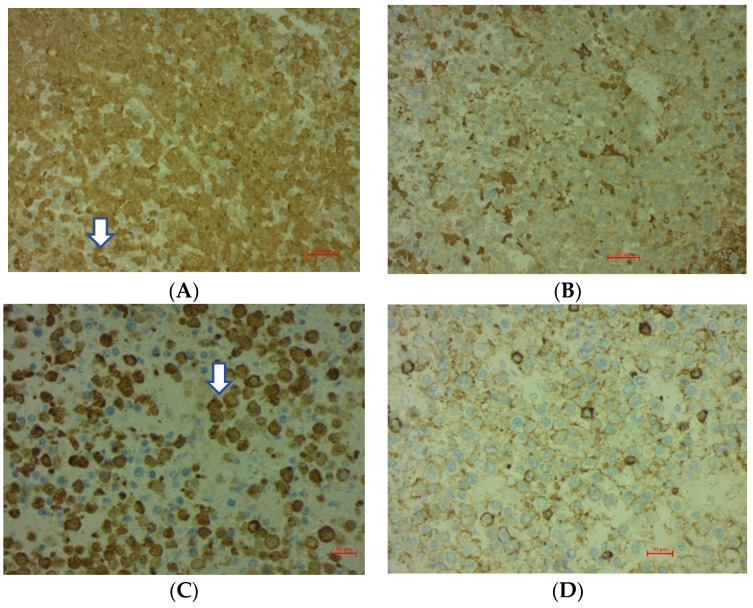
(**A**) kappa light chain immunoglobulin shows cytoplasmic expression (arrow); (**B**) a lack of the lambda immunoglobulin light chain. (**C**) Immunohistochemical staining for CD138 shows strong cytoplasmic positivity (arrow). (**D**) LCA.

**Figure 6 brainsci-12-00705-f006:**
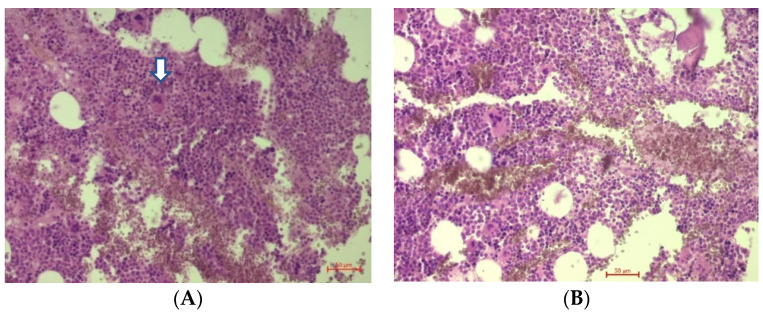
(**A**,**B**) Bone marrow at 20× magnification. Bone marrow was normocellular, haematopoiesis is maintained, small–medium-sized megakaryocytes (arrow).

## Data Availability

Not applicable.

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
