# Peer review of "Intracerebral Plasmacytoma in a Patient with HIV-1 Infection and SARS-CoV-2 Superinfection"

_brainsci, 2022, doi:10.3390/brainsci12060705_

Round 1

Reviewer 1 Report

The single case study shown indicates the complications in diagnosis when multiple infections occur in HIV patients. The baselines are not very clear, the authors need to justify why this study is important with respect to the literature. As such it is previously known that COVID19 and HIV can make things complicated - you need to bring in several literature on this aspect to compare and benchmark on what you have observed.

The analysis is overall a bit weak to suggest the correlations between the various infections. Its not clear which particular infection complicates the treatment and diagnosis. Given there are already, MRI images, it will be good to perform a detailed analysis, including making use of image analysis and statistical analysis techniques to arrive at an early conclusion. 

Author Response

The case we described is rare and there are few available literature sources that describe intracerebral plasmocytoma. We wanted to point out the difficulties in diagnosis and treatment. In our case, the diagnostic difficulties were exacerbated by the patient's infection with SARS CoV-2 infection. The patient was diagnosed with HIV in 2020, shortly before the plasmocytoma was discovered.
Our case report also reveals diagnostic and treatment difficulties caused by other factors. It is also important to note that there is a differential diagnosis of other similar diseases.

Thank you. Your suggestions will be corrected. 

Reviewer 2 Report

The authors present an interesting case of a patient with an extramedullary manifestation of plasmacytoma. The patient contracted HIV-1 and SARS-CoV-2 infection before developing a malignant neoplastic lesion in the right basal ganglia area with close association with the lateral ventricle. This is an extremely rare case. Therefore, this report warrants publication. However, there are several aspects that need to be addressed before being published in Brain Sciences.

  1. In the abstract, the sentence at the beginning of the chapter, “Clinical management and autopsy results of the case are described,” is repeated at the end. Please revise.
  2. The introduction contains information about HIV infection in Latvia, including the incidence in risk groups and the definition of late presenters. This does not add valuable information to the case report and should be shortened.
  3. In the result section, it is mentioned that the CSF studies showed “cytosis,” which is defined as transport mechanisms out of the cell. What exactly does that indicate in this context?
  4. As reported in the result section, the patient was admitted to neurosurgery to obtain a biopsy (pg. 3, ln 110). The fact that this biopsy never happened is reported on pg.4, ln 313. It would be helpful if these two aspects were reported together. It is somewhat surprising that the risk for a biopsy was considered too high in a young female patient with a space-occupying lesion in the brain requiring urgent treatment. It would be interesting to know more about the reasoning leading to that decision.
  5. Was the CSF sample obtained by lumbar puncture? How was the presence of CD138+ plasma cells interpreted? Were there any atypical/lymphoblastic cells detected in the CSF sample?
  6. Obviously, the patient was readmitted in September 2020. What was the treatment between March and September 2020? The steroid – treatment was initiated in September 2020, so no treatment between March and September 2020?
  7. If no treatment was admitted in March, what can be the reason for the partial response in September 2020? Please elaborate on the course of treatment between March and September 2020.
  8. On pg 7, ln 168, it is mentioned that the “Plasmacytoma 167 was aggressive because first time on MRI it was not diagnosed”; however, on page 2, ln 68, a CT scan is mentioned, not an MRI. The initial MRI showing the intraaxial tumor was obtained 14 days after initial admission when the patient developed a hemiparesis. Why is it that 1.5 months are reported as a time period in the discussion? Please elaborate.
  9. In conclusion, the diagnosis of solitary plasmocytoma is based on the immunoglobulin light chain clonality analysis. How was the differential diagnosis B – cell lymphoma ruled out in the absence of Bence Jones proteins and paraproteins in the serum electrophoresis?
  10. Given that this patient was infected with HIV, were there any diagnostics of Epstein-Barr virus affection performed? EBV shows frequent association with immunodeficiency due to HIV/AIDS and also with multiple myeloma as well as
  11. There are several typing errors and linguistic mistakes - professional language editing is necessary!

Round 2

Reviewer 1 Report

As a research paper the analysis is weak. However, for a case study, indicating a unique case, sufficient analysis is provided. The lack of reference cases makes it a challenging study to draw comparisons. Authors can however, do a comparison with cases that do not have HIV to draw further conclusions on co-infections. There other reported cases on co-infections and co-morbidilities, which can be a basis for comparison. There are previous works discussing the neurological impact of COVID, which can be useful for base the work in the study.

https://pubmed.ncbi.nlm.nih.gov/32683890/

https://pubmed.ncbi.nlm.nih.gov/33446327/

https://pubmed.ncbi.nlm.nih.gov/33464535/

https://pubmed.ncbi.nlm.nih.gov/34169443/

I would suggest the authors to strengthen the background part of the work. Include, a bit more details analysis of background and literature on the case. The imaging interpretations needs citations. The MRI images will go better with further contrast corrections, and highlight the regions of interest, and label them. Same goes with other images used.

Author Response

Thank you for review.

We added citations and corrected images with arrows. 

We compiled other articles and drew conclusions, which we summarized and added to the article (189-200)

Reviewer 2 Report

The authors have significantly improved the quality of the paper. In addition, all aspects of my previous review have been addressed. 

Author Response

Thank you for review.